# Broadening participation in science through arts-facilitated experiences at a cultural festival

Mark S. Rosin[1,2☯]*, Martin Storksdieck[3☯], Kari O'Connell[3☯], Brianna Keys[3], Kelly Hoke[3], Bruce V. Lewenstein[4]

**1** Department of Math and Science, Pratt Institute, Brooklyn, New York, United States of America, **2** Guerilla Science LLC, Brooklyn, New York, United States of America, **3** STEM Research Center, Oregon State University, Cornvallis, Oregon, United States of America, **4** Department of Communication, Cornell University, Ithaca, New York, United States of America

☯ These authors contributed equally to this work.
* mrosin@pratt.edu

**Data Availability Statement:** The data set on which this research is based is now publicly available at the following URL, hosted by the

## Abstract

A key broadening participation challenge for science communication is to reach non-traditional audiences beyond those already interested in science. In this study we test a "Guerilla Science" approach that blends elements of *access*, by removing barriers to participation, with those of *inclusion*, by designing participant-centered activities, for reaching an art-interested adult audience at the FIGMENT art festival in New York City. Our results show that participants at Guerilla Science were similar to festival goers overall in their connection to and interest in science, indicating the effectiveness of the approach for engaging non-traditional audiences and those who might not self-select into science activities.

## 1.Introduction

### 1.1. Broadening participation in science and the self-selection bias

A key challenge for informal learning (ISL) activities is to reach audiences who do not self-select as "science interested;" That is, audiences who do not actively seek out science learning opportunities. This challenge is motivated by two principals: First, a belief that learning science (defined as science, technology, engineering, and math: STEM) is critical for full participation in modern society leading to a commitment by educators to increase the number of people with access to science; and second, a recognition that some people are actively excluded from STEM learning through broad social structures of inequity and lack of diversity. Addressing issues of *access* and *inclusion* have been identified as critical for broadening participation [1, 2]. Providing equitable access means that no one is systematically excluded based on race/ethnicity, gender, sexual orientation, zip code, socioeconomic status and other personal characteristics unrelated to science engagement itself [3]. Beyond access, however, inclusion demands experience designs that respond to the personal context of the full diversity of potential participants. Access requires tailored support for different

Scholars Archive at Oregon State University:
https://doi.org/10.7267/hx11xq17s

**Funding:** This material is based on support from
the National Science Foundation (Award #1612719,
M.R., https://www.nsf.gov/) and the Simons
Foundation (Award #415600, M.R., https://www.
simonsfoundation.org/). The funders had no role in
study design, data collection and analysis, decision
to publish, or preparation of the manuscript.

**Competing interests:** We have read the journal's
policy and the authors of this manuscript have the
following competing interests: Mark Rosin is a part
owner of Guerilla Science Global and sits on its
Board of Directors. This does not alter our
adherence to PLOS ONE policies on sharing data
and materials. The remaining authors declare that
there are no conflicts of interest

participants, whereas inclusion requires a culturally-aligned approach to the event's choice
of topic, as well as its design.

Many sites and activities that are traditionally the focus of ISL–such as science centers,
natural history museums, planetariums, and science festivals–engage primarily with sci-
ence-interested publics. But many previous studies have shown that they mostly attract an
audience that is not representative of the population at large [4–9]. Non-visitors might not
simply lack interest in science, but more importantly may feel alienated by ISL settings for
their cultural expectations, or might be limited by cost and location [10, 11]. Consequently,
ISL settings suffer from "self-selection" bias, attracting the "usual suspects" of those with
means to engage in free-choice STEM activities and who are sufficiently science-interested
to choose those settings as part of their spare-time activities. In doing so, many ISL settings
fail to attract those who do not actively seek out science learning experiences or do not feel
welcomed and included in the spaces and formats that dominate ISL settings. Thus the
problem grows: those already invited into science get more opportunities, while those who
feel excluded have fewer and fewer opportunities [10, 12]—what historian of science Marga-
ret Rossiter [13] called the "Matilda effect".

To meet the need of individuals who would not ordinarily feel welcomed or comfortable in
science-focused settings, a growing number of science centers, museums, and funding agen-
cies have turned to integrated arts and science programs. The rationale for this focus is that
integrated art-based learning offers an experiential and interdisciplinary approach to STEM
education, one with a distinctive new set of tools to advance creativity and engagement among
learners [14–16].

However, even this approach—that of combining science and art, or more broadly, science
and culture—can lead to the same self-selection process [17]. A 2013 meeting at MIT on
"Evolving Culture of Science Engagement" brought together science communication practi-
tioners with social science and education researchers and evaluators to explore new modes for
engaging the public with science (http://www.cultureofscienceengagement.net). One of the
key messages that emerged from that meeting was that traditional forms of science communi-
cation as part of popular culture have focused too strongly on the "science" end of the cultural
spectrum, to the cost of broadening participation agendas. The 2013 meeting suggested a dif-
ferent way of approaching the issue: Culture exists as an ever-changing mix of themes and
ideas, often mediated through the arts, music, theater, and so on. Consequently, for many
potential audiences, science would be most attractive when embedded into those aspects of
culture that they are most interested in, or feel most connected to. To engage them with sci-
ence, hence, suggests that we embed science, or make science part of, the culture of which they
are part.

For example, the context of Guerilla Science–a particular intervention that is the subject of
this study–the associated Theory of Change describing *how* science can be made part of cul-
ture, and the associated impact, is as follows [18]:

Activities: Live creative public science experiences in cultural and community spaces.

Outputs: People who would not otherwise engage with science build a connection with it.

Outcomes: People's lives are enriched through meaningful and relevant relationships with
science.

Indeed, in an earlier study [19], again, focusing on Guerilla Science [18], we demonstrated
that embedding science into culture (as opposed to the inverse), can broaden participation,
albeit in one specific case. That study focused on participants at an expensive and self-

contained music and art festival held at an isolated rural location in the US state of Oregon. Though the case addressed inclusiveness, challenges for access were still present. Furthermore, that festival was marketed and took place during the total solar eclipse that was visible across the USA in August 2017 and which became a major media event, garnering enormous situational attention and interest from a large portion of the population, both science-affine and non-science affine [20, 21]. In this paper, we therefore consider a more complex case of embedding science experiences into a family-friendly cultural festival held in New York City with no admission costs and easy access for a vast number of people. Because these two festivals occupy radically different positions in the live-event science learning space—a large and important component of the informal science learning landscape [22]—this study provides an essential piece of additional evidence that supports our initial conclusion. That is, it provides us with the opportunity to expand upon and test our earlier results; an important goal, given the well-known replicability problems throughout the social sciences.

## 2. Methods and framework

Our research sought to investigate several questions:

1. Can participation in science or STEM be broadened by providing science or STEM engagement opportunities in public spaces as unplanned encounters?

2. Do such "pop-up" or "stealth approaches" for science or STEM engagement primarily select for individuals who already have an above-average affinity towards STEM?

3. What are the outcomes of participation in these events?

The answers to these questions allow us to make causal inferences as to whether arts-based informal science learning activities, at least as exemplified here, have the potential to broaden participation in science, or whether such activities will mostly attract the science-interested subpopulation who happens to attend.

To operationalize these research questions, this study uses the context of a Guerilla Science event, Sensorium, at the FIGMENT Festival. FIGMENT is self-described as a "free participatory arts event that celebrates creativity by challenging artists and participants to find new ways to create, share, and dream," the festival hosts a rich array of arts, culture and educational programs as well as expansive open spaces and historic buildings on Governors Island, New York, over the weekend of June 23rd and 24th in 2018 [23]. On the other hand, Guerilla Science creates in-person events that bring scientists and members of the public into face-to-face contact within the context of immersive storyworlds, that is, science activities that are embedded in broader stories or narratives that provide cultural context and meaning [18, 24]. The Guerilla Science events occur in the places and spaces where science is least expected—for example, at cultural, music, and art festivals, unclaimed urban spaces, social clubs and other locations which by design visitors find accessible and that are generally not associated with science or STEM encounters. It is the "stealth" element that supports the *access* for new audiences, and the choice of cultural context, festivals in this case, that supports *inclusion*.

Implicit in our methods is a conceptual framework that understands festivals as instances of socio-cultural production. Specifically, festivals are temporary, ephemeral spaces, and that it is this property that lends them potency as vehicles for learning, individual, and social changes [25]. Festivals can act as emancipatory spaces where participants can let off steam, explore, and learn through play. In particular, for many participants, they are spaces for "experimentation with identity and the articulation of identity politics that may often be less feasible and acceptable–and in some cases socially circumscribed–in everyday settings" [26]. Their contemporary

form is most readily traceable to the countercultural movement of the 1960s and beyond: Woodstock, Free Festivals, and Glastonbury. While it is undoubtedly true that festivals are commercial entities, constrained by market forces, visitor numbers, and supported by entire fields of festival "tourism" and festival "event management" [25], part of their appeal remains deeply rooted in their countercultural carnivalesque origins. Therefore, even in their most commercial realizations, most festivals are still places where social norms can be critiqued or rejected; alternative ways of living and being can be imagined and experimented with; and social issues can be promoted [27]. Thus we anticipate that festivals may address the issue of inclusiveness, by representing opportunities for people of multiple identities to connect with a wide range of activities.

The origins of this understanding of festivals is most readily attributed to the early 20th Century Russian semiotician Mikhail Bakhtin [28, 29], whose model of carnival can be identified in many contemporary festivals.

We also note that the emancipatory role of festivals is generally distinct from, although not entirely so in this specific case, the emancipatory role of [informal] science education, as seen through the lenses of Dewey and Friere, see e.g. [30].

## 2.1 Setting

Governors Island is a 172 acre island in the heart of New York City's harbor, a five minute public ferry ride away from Brooklyn and Lower Manhattan. It hosts a rich array of arts, culture and educational programs—nearly 70 free exhibitions, installations, performances and festivals in 2018—as well as expansive open spaces and historic buildings. The island is a popular seasonal destination for locals and tourists during the summer months with approximately 750,000 visitors over the summer months of 2018. Of those visitors, 75% came from New York City, and an additional 13% from the metro area. Visitors represent 90% of the city's zip codes, with no more than 3% from any one [31].

On June 23rd and 24th 2018, Guerilla Science transformed a disused officer's quarters and the surrounding lawn space into the "Sensorium," an otherworldly exploration of how the human body works inside and out (Fig 1). Sensorium, in its marketing materials, is described as "a sensory adventure that explores the most complex thing in the universe: us. Visitors will hear tastes, feel sounds, and see the shape of smells to uncover some of the mysteries of how our bodies make us who we are." The event thereby drew on the carnivalesque practices of centering the body and bodily functions (albeit in a manner less grotesque and scatalogical than in many historical realizations).

Sensorium's weekend-long program included 20 installations and performances distributed across 10 rooms in the historic house and adjacent lawn space as part of the FIGMENT Festival. Visitors created bespoke jewelry from their own bacteria at a Precious Germs Emporium, met with a plastics life coach at the Plasticization Clinic, and entered a room where everyone appears to be the same height. There was also a collaborative room-sized brain-art sculpture, a touch and sound lab, and an experimental device that translates smells for the olfactorily challenged. Festival-goers were able to enjoy an auditory tour of the mind, twist their sense of taste at the Flavor Feast (Fig 2), and discover what happens to the body in outer space at an intergalactic yoga studio. The works on display were developed through a scientist and artist residency program hosted at the Pratt Institute and staffed by their creators (Fig 3), and taken from a back-catalog of Guerilla Science events that had been developed iteratively over the last decade and staffed by volunteer scientists and informal science learning educators.

The FIGMENT Festival was designed to attract an audience focused on music, art, being outside, and alternative culture. For the audience at FIGMENT festival, *accessible* meant that

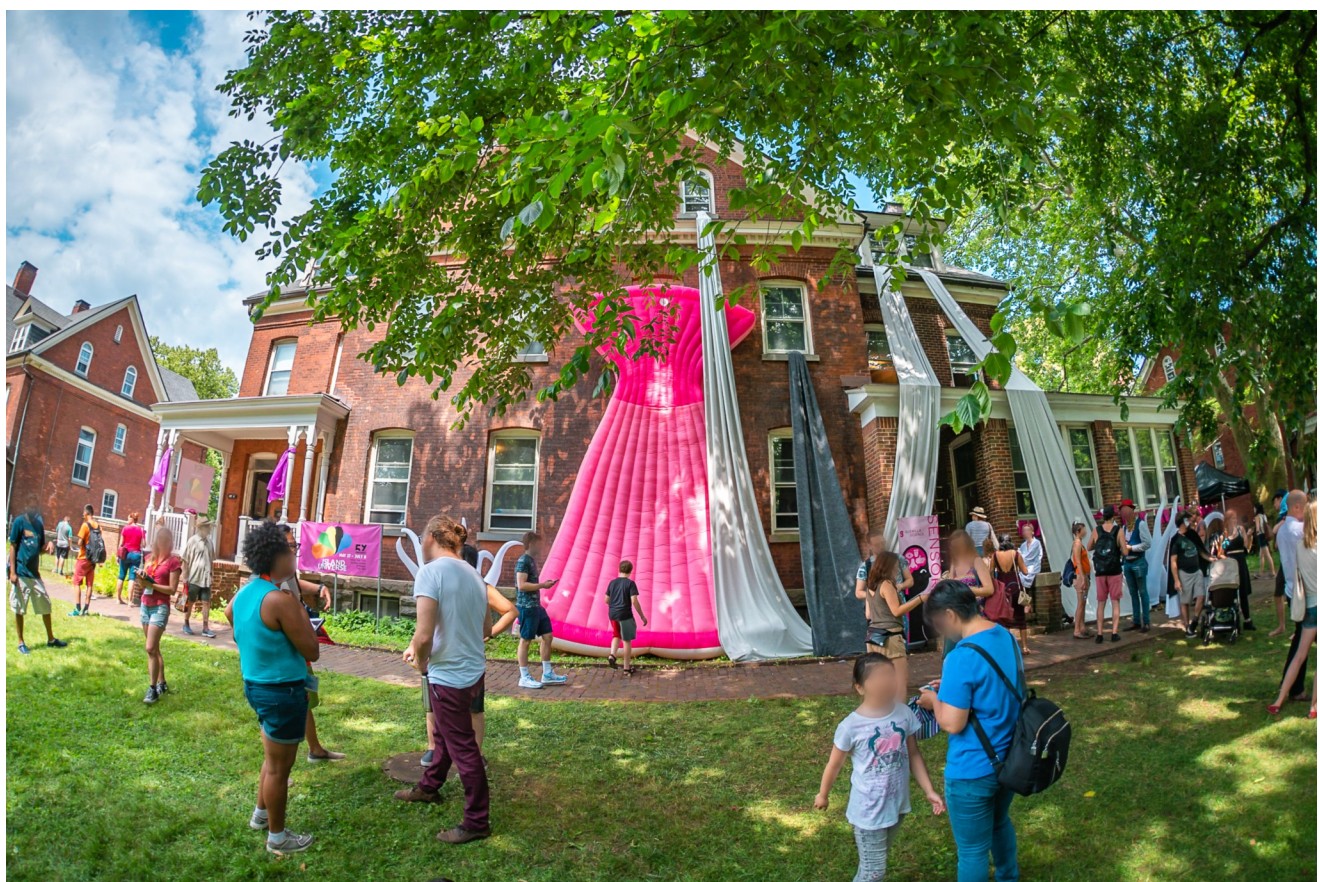

**Fig 1. Sensorium, a multi-room installation in a disused officers' quarters and on the surrounding lawn exploring the human body.** The event took place one summer weekend in 2018 as part of the FIGMENT Festival.

there were few barriers to entry *to the festival*, i.e. the festival and the ferry ride to the island were free and located near public transit hubs, so once visitors were at Governors Island, the impediments to participation were minimal. *Inclusion* meant, at least in part, creating modes of engagement that spoke to the likely interests of festival visitors centered on music and art, among others.

Activities at Sensorium generally took the form of hands-on, open-ended, learner-centric experiences. The science-learning goals of these activities were centered on promoting science as a way of knowing, as a process, and a set of concepts; and to inducing joy and excitement in participants as a means to helping them develop identities as science learners [4]. For example, in Anatomical Life Drawing, audience members gained an alternative perspective on the human body. In this unique drawing class, the internal anatomy was painted onto a semi-nude model before the event, which was then replicated in the audience's life drawings. Participants were invited to engage in the epistemic practices of noticing and questioning the functions and form of circulatory system, from both a scientific and aesthetic perspective [32]. The event was co-hosted by an artist who painted the model's body in advance and provided drawing tips during the event, and a medical professional who explained the underlying anatomy—the event was designed to feel like a conversation between the two, and was informal, friendly and open to questions (Figs 4 and 5).

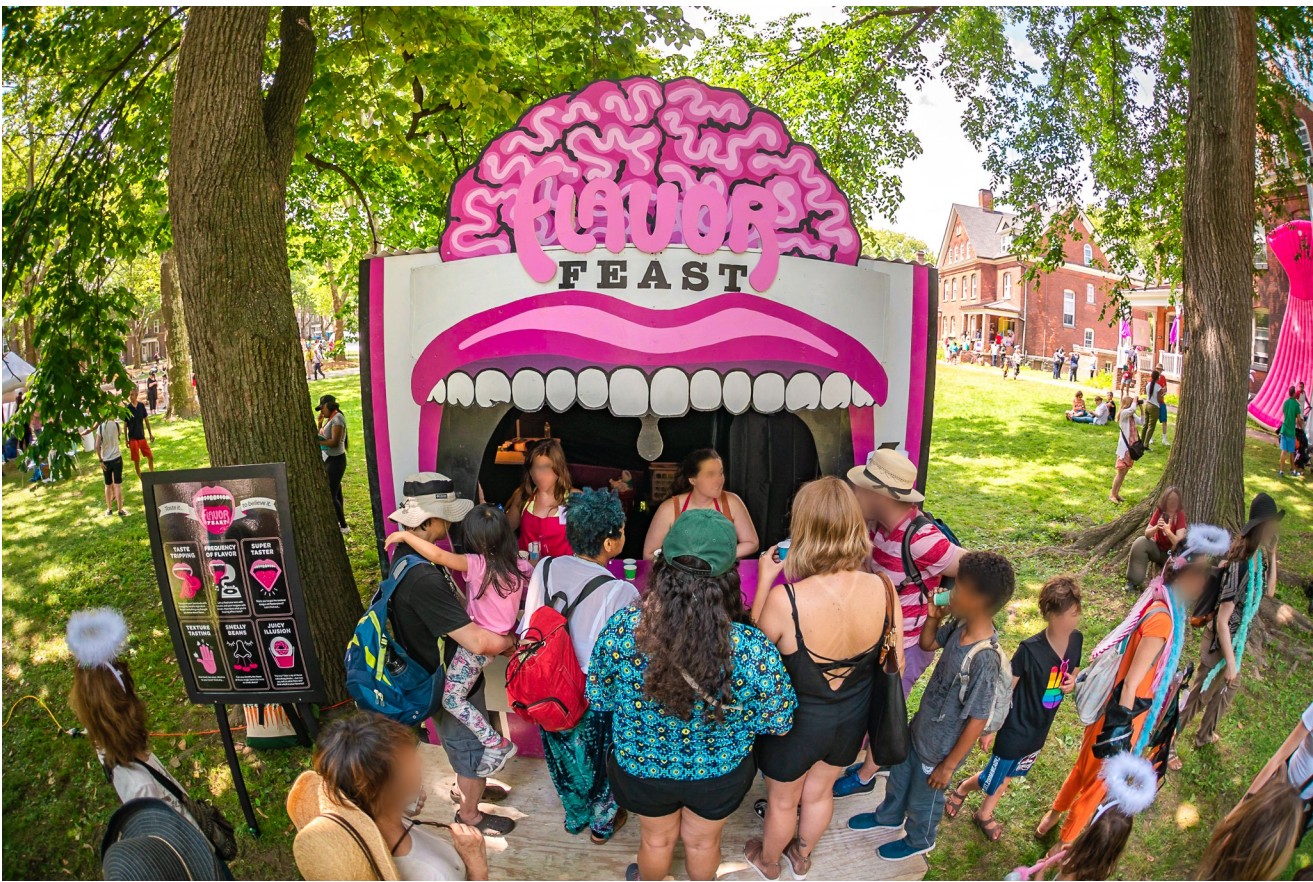

**Fig 2. Flavor Feast, a pop-up food stand featured as part of Sensorium.** Participants took part in a series of edible experiments that show how smell, touch, sight and hearing can influence taste.

## 2.2 Research design and data collection

To answer our research questions (section 2) we sampled from the general Governors Island visitor population at various locations on the island, including areas associated with the FIGMENT festival, and from the population participating in Sensorium events. We collected data using multiple methods for the purpose of methodological triangulation [33]. We employed short entry and exit interviews at the officer's quarters that were part of Sensorium and the surrounding exhibits, post-event paper-pencil feedback forms available after each scheduled event and in between rooms in the officer's quarters, and short comparison interviews for visitors to Governors Island who did not attend Guerilla Science events. Each instrument included both open-ended and closed-ended questions to gather a robust set of data (Table 1).

We collected data on June 23 and 24, 2018. The data collection team included four researchers and 18 graduate students who were selected for their previous experience with social science data collection. These students were given training before the FIGMENT Festival. We intercepted potential respondents for interviews by strategically placing team members around the officers' quarters and the surrounding lawn exhibits. Interviewers were asked to use a saturation approach and intercept as many individuals as possible for the entry and exit interviews and post-event feedback forms. Interviewers intercepted individuals for general festival population (comparison) interviews in other areas of Governors Island away from Sensorium and exhibits; additional comparison interviews were conducted within the area of the FIGMENT

# 2018 RESIDENT ARTISTS

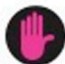

### PRECIOUS GERMS
Alexandria Palaferri Schieber

*Explore gut-microbial interactions through first-hand sensations of what it's like to be a germ, and play to win an original microbial art piece.*

Discover the complexities and simplicities of bacteria at this shop, where we invite customers to become one with the billions of precious germs in your gut.

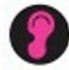

### THE WOMB OF THE WORLD
A Feeling Sound Spa
being(())sound
+ Dr. Claudia Aguirre

*Touch is the first sense we develop and is deeply connected to vibration and sound. Experience hearing in a new way using this tactile sense.*

Can you relearn to listen with your body? Can the sensations of sound bring us together? What would it feel like to experience sound as the touch of the world?

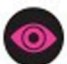

### BALLOON BRAIN
Sarah Vitak

*Try walking through a jungle of neurons, build new parts of the brain, or make memories by moving, disconnecting, and reconnecting brain cells.*

Join us in an immersive interactive brain section made entirely of balloons, and help recreate the connections inside this fascinating organ.

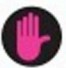

### THE FABRIC OF SPACE-TIME
Kayla Lewis + Gabriela Lemos

*Will you be clothed in loop quantum gravity, generalized string theory or causal sets? Find out in this exclusive inventory of quantum looks.*

Which theory of quantum gravity most defines your style? Peruse cosmic clothing lines, choose your favorite weaving of space-time to wear, and uncover the fabric of everything.

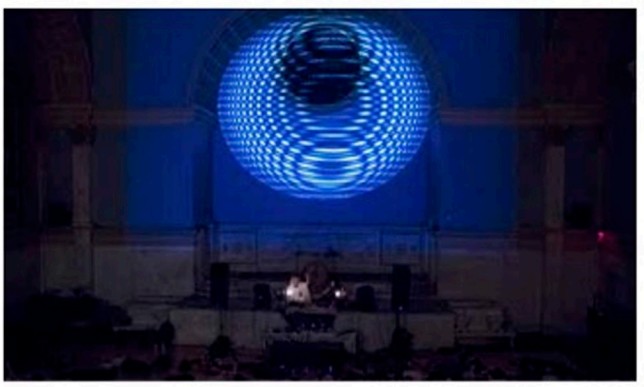

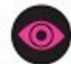

### PLASTICIZATION CLINIC
Lauren Ruiz

*Come see if we can discover our own BPA levels, find out the future of plastics, and uncover the effects of plastic waste on the planet and our everyday lives.*

How plastic are you? Determine the plastic content of your life and body with a specially trained plasticization consultant.

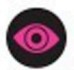

### BEST BIAS
Thomas Koff

*Find truth in our rhyme-as-reason refrigerator sets, see faces in our newest pareidolia TVs, and throw on platform shoes to become the same height as new friends.*

Delve into the biases of your brain in our concept kitchen, as we launch a product line of household appliances to correct outdated biases.

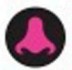

### SMELL-X
Olivia Jezler

*Special agents will seek out persons with a very special gene to become smell translators and compete to bring smell back to humanity.*

What if you lost your sense of smell? We will transport you to a world without scent to learn how we can understand smell through other senses.

**Fig 3. A page from the Sensorium program describes the novel installations and activities created by artists and scientists working together.** The installations and activities were designed to appeal to attendees' artistic identities, while supporting STEM learning.

festival itself, but away from the Sensorium. The comparison sample was intended to represent individuals who visited Governor's Island that day, and who could have opted to attend Guerilla Science activities at the Sensorium. Interviewers were asked to approach people at random and invite them to complete an interview. To facilitate a diversity of respondents, interviewers were further instructed that they should only interview one individual in any groups they approached. Interviewers gave respondents a laminated copy of the interview questions to ease response burden and interviewers transcribed responses on an iPad. Respondents who completed the comparison spot interview were given a Guerilla Science pin as an incentive for participation. Paper feedback forms were handed out to participants in the scheduled events and gathered by one of the data collectors.

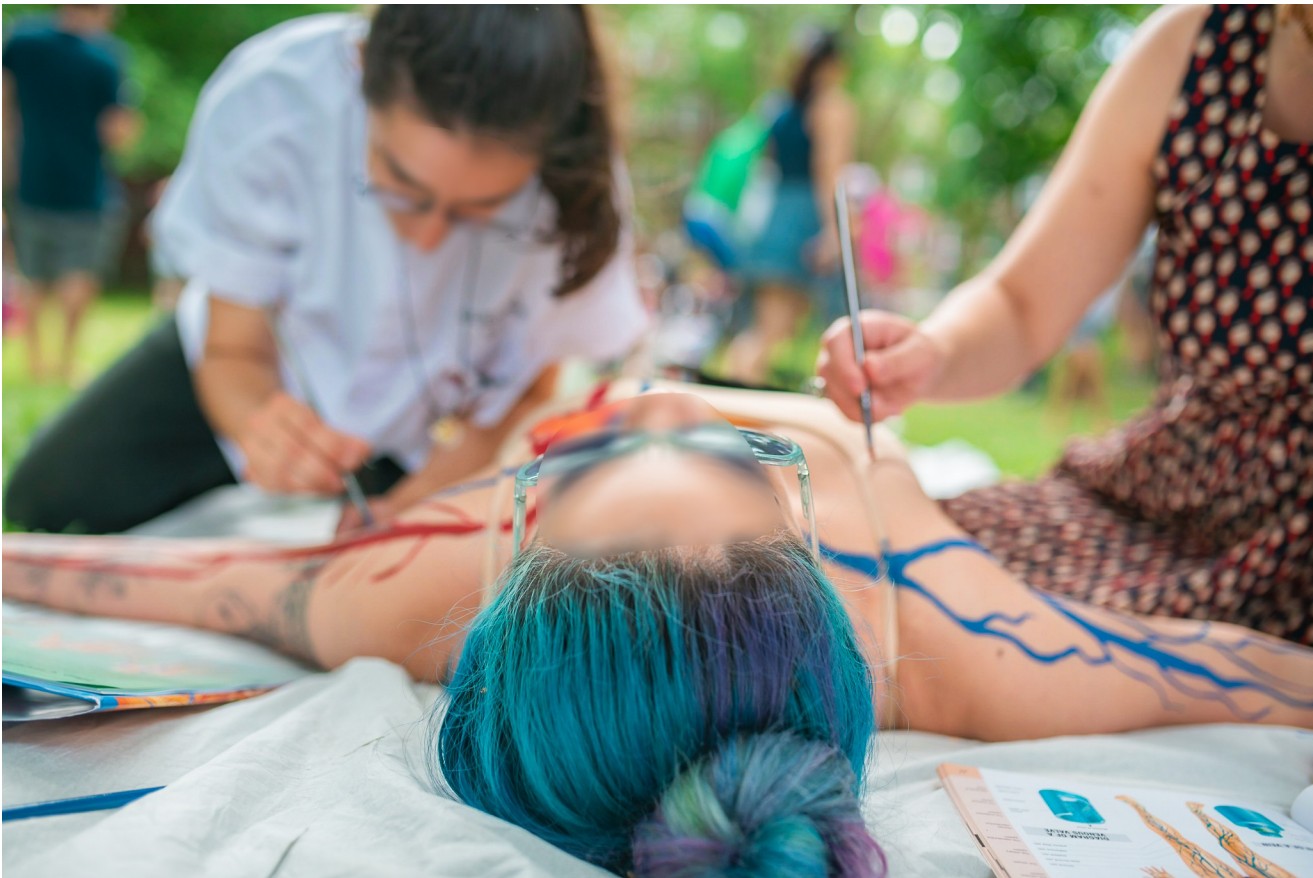

**Fig 4. An artist's model at Anatomical Life Drawing.** Here the anatomically accurate veins and arteries are being painted directly onto the model's skin for the benefit of audience members (Fig 5).

The Oregon State University (OSU) Institutional Review Board (IRB) approved the study as an exempt study with a waiver of documentation (signature) of informed consent. For paper surveys, we included brief study information and consent language at the top of the survey, and participants gave their consent by filling out part or all of the paper survey and turning it into the box (labeled as such). For interviews, we read study information to potential participants before starting the interview, allowing for verbal consent. Our study only included adults (18+), and as part of the verbal consent we included the requirement to be 18 to be part of the study. The underlying data on which our results have been made available in a publicly accessible archive [34].

In contrast to data collection for a previous similar study at the Oregon Eclipse Festival (Bisbee O'Connell et al., 2020), a camping-based festival, which occurred over multiple days and therefore created the challenge of repeat visitation and repeat participation in the study, fewer individuals were likely to attend FIGMENT on both days, or repeat participation in Sensorium. Nonetheless, data collectors asked intercepted individuals about prior participation in Sensorium or in the current study before proceeding with interviews.

## 2.3 Development of instruments

We started this study with the same instruments used in Bisbee O'Connell et al. [19] that focused on the Oregon Eclipse Festival, and made changes based on lessons learned from that

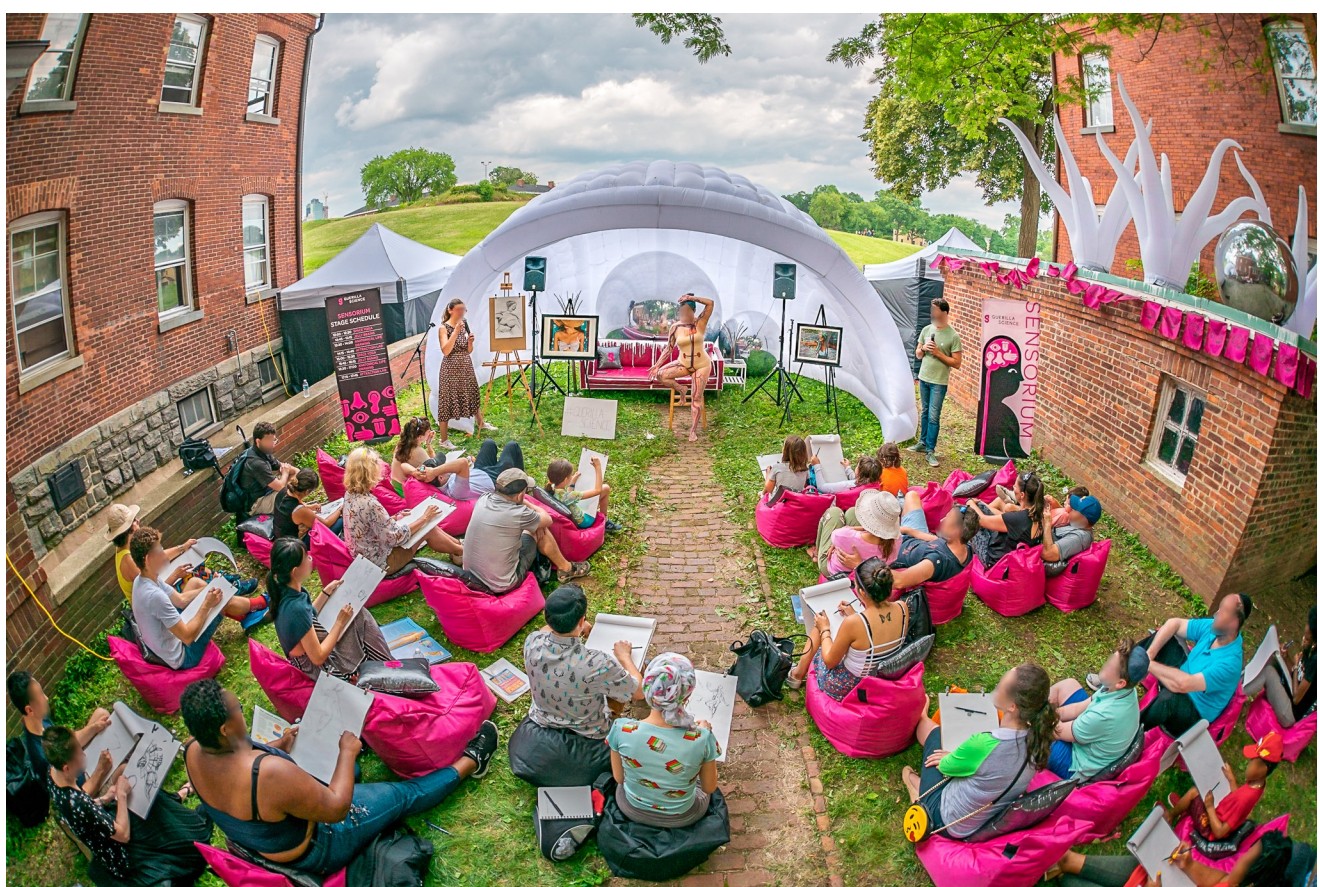

**Fig 5. Audience members drawing at Anatomical Life Drawing, an art class with a difference in which audience members engage in a basic model sketching exercise and are introduced to elements of human blood circulation.** The event took place on the Sensorium stage, on the lawn next to the officers' quarters.

experience. The original data collection instruments were developed for the unique festival context with the aim of studying Guerilla Science events. In developing the original instruments we leaned heavily on the recommendations of Bultitude and Sardo [35] and Sardo and Grand [36] for keeping survey questionnaires and interviews as short as possible to lessen the interruption of the participants' festival experience. We also opted for informal and conversational intercept styles to stay within the norms and expectations associated with a family-friendly, art-oriented festival. We developed close-ended questions about the motivations and expectations for attendance, and perceptions and evaluation of the event based on open-ended responses from Bisbee O'Connell et al. [19]. For this study, we supplemented the original feedback forms with additional questions around general participation in arts and science-oriented spare-time activities based on items from the Science and Engineering Indicators survey project [8] in order to render our data more directly comparable to national indicators of views of, and engagement with, science or STEM. Throughout we matched items and open-ended questions with constructs (Table 1).

In conducting our statistical analysis, we ignore the possibility of any systematic bias in how participants interpret the term science, be it in a narrow topic-based sense or an expansive sense to include activities such as cooking or gardening. Earlier work has suggested that

**Table 1. Overview of instruments and population from which they sampled.**

| Instrument (n) | Population | Constructs |
|---|---|---|
| Entry Spot Interview (n = 227) | Guerilla Science participants, before entry into Sensorium | • Views on science<br>• Engagement with science<br>•Motivation and expectations for attendance<br>• Participation in arts and science spare-time activities |
| Exit Spot Interview (n = 326) | Guerilla Science participants, after events, as they exited Sensorium | • Perceptions and evaluation of event<br>• Takeaways from participation<br>• Intentions for future behaviors<br>• Motivation for attendance*<br>• Views on science*<br>• Engagement with science* |
| Feedback Forms (n = 202) | Guerilla Science participants, immediately after events, as they exited events in the house or as they exited the stage area | • Perceptions and evaluation of event<br>• Takeaways from participation |
| Comparison Interview (n = 484)[1] | Governors Island visitors | • Views on science<br>• Engagement with science<br>• Motivation for visiting Governors Island<br>• Knowledge of, attendance, interest in Guerilla Science<br>• Participation in arts and science spare-time activities |

[1] Total number of people who started the interview. 282 stated they came for FIGMENT, and 132 were aware of the Guerilla Science Sensorium (but had not attended yet).

* Not asked if an entry interview was conducted with the respondent (questions were included in entry interview).

labeling an activity as "science" may have to be explicit for participants to change how they self-report their engagement with science [37].

## 2.4 Data analysis

We used SPSS 25 to perform analyses of closed-ended data and reliability analysis for the views on science and engagement Likert-type scale items.

To portray the audience in terms of their relationship to science, we employed audience segmentation as an analysis tool. We allowed discrete audience groups, or audience segments, to emerge from the combined score of multiple items measuring respondents' views on and engagement with science-related issues; the exact measures for this segmentation were derived from Bisbee O'Connell et al. [19]. Responses for this segmentation included choosing (or not) science as 'a topic that best describes them and their interests', responses to four Likert-type scale items, and describing (or not) science as 'valuable', 'fascinating,' and 'fun'.

The closed-ended survey items produced nominal and ordinal data. We used Chi-square tests of independence to investigate the association between variables, including the relationship between the audience segmentation and participation in particular data collection instruments.

# 3. Results

## 3.1 Why did people participate, who decided, and what did they expect?

Guerilla Science creates novel experiences that draw from both science and a variety of artistic disciplines. Understanding whether this hybrid form of public engagement is able to attract and retain new audiences to science requires us to first understand motivations for participating and the decision process towards participation. The key question is the degree to which participation was deliberate and based on reasonable deductions about the nature of the experience. Ultimately, the decision to engage with Guerilla Science events allows us to estimate the degree to which stealth science approaches are impactful. Consequently, we inquired with people as they entered Sensorium about their reasons for participating, their expectations for the upcoming experience, and what they believed the outcomes from the experience might be for them.

Almost all individuals who participated and were intercepted for interviews or surveys were part of a group, which raises the question whether they themselves intended to, or decided to, participate in FIGMENT and Sensorium. The majority of individuals who entered Sensorium stated that they decided to attend FIGMENT and decided to participate in Sensorium, (61% and 69%, respectively) vs. 30% who tagged along with someone else who wanted to attend. Guerilla Science was a new cultural phenomenon for the vast majority of those who participated in Sensorium. In entry interviews (n = 227), 86% of the respondents reported that they had not heard of Guerilla Science before (i.e. were not familiar with Guerilla Science as a concept or the organization). Similarly, most respondents (71% of 452 who answered the question across instruments) who had not (yet) attended Sensorium were unaware that FIGMENT featured a set of events and activities under the moniker of Guerilla Science. That is, most visitors to Governor's Island were not aware of Sensorium as a program element for FIGMENT, and most of those who then attended Sensorium as part of FIGMENT were likely not familiar with Guerilla Science as a concept or organization. This result is important since it indicates that Guerilla Science patrons at FIGMENT were not seeking Guerilla Science-style experiences, or were not Guerilla Science enthusiasts, even though the organization had disseminated widely throughout its New York network that it was present at FIGMENT.

When choosing from a close-ended list about motivations for participating in Sensorium, on average, respondents chose 2.4 of 6 answer options, showing the complexity of motivations for visiting. The dominant reason for attending was to experience something new, which was chosen by 56% of respondents. The picture that emerges is that Sensorium attendees wanted to experience something novel with friends or family that was culturally enriching (Table 2).

Curiosity, novelty, and entertainment were also the dominant expectations for Sensorium (Table 3). Respondents chose on average 1.5 of 6 provided answer options from a list of possible expectations for their visit. Only 15% of those who entered Sensorium wanted to have a

**Table 2. Reasons for attending Sensorium.**

|  | *Percent of respondents (n = 227)* |
|---|---|
| To experience something new | 56% |
| To have a social experience | 44% |
| To explore culture or increase knowledge | 37% |
| To spend time with family | 37% |
| To relax or refresh | 31% |
| To escape routine | 28% |
| Tagged along | 10% |

**Table 3. Expectations for visiting Sensorium.**

|  | *Percent Respondents (n = 227)* |
| --- | --- |
| I am simply curious | 37% |
| I want to experience something new or different | 31% |
| I want to enjoy myself | 29% |
| I want to learn something | 22% |
| I want to have an art-related experience | 20% |
| I want to have a science-related experience | 15% |

science-related experience, though similarly, just 20% opted for an art-related experience. More than a third were simply curious (which we interpret as having no expectations), and another 31% wanted to experience something novel, which means that an important driver for the visit was exploration of the unknown. Fewer than a third opted for enjoyment as a driver, indicating that most visitors did not attend from a pure entertainment perspective. In fact, more than a fifth of the visitors expected to learn something during the Sensorium experience.

## 3.2 Did Guerilla Science's Sensorium attract an audience that self-selected for science from within the visitors of FIGMENT and Governor's Island?

In order to estimate whether Sensorium attracted an audience with a strong relationship to science out of the overall FIGMENT population, we compared the 373 participants of Sensorium with a comparison group of 466 FIGMENT visitors who had not visited Sensorium at the time of interception. The Sensorium data represent entry and exit interviews to the Sensorium and might therefore include a situational positive effect on science enthusiasm due to participation in Sensorium activities. We divided the populations into five audience segments, ranging from 'science enthusiasts' to 'disconnected from science' and present results from this study (Fig 6) and equivalent ones from the previous research at the Eclipse Festival in Oregon (Fig 7). The data show that the Sensorium participants came from the full range of audience segments, from science-disconnected to science enthusiasts. In fact, Sensorium participants were more likely than FIGMENT attendees to belong to the science disconnected or uninterested audience segments (a combined 64% for Sensorium versus a combined 45% for FIGMENT as a whole). Conversely, more than a fifth of the FIGMENT comparison group fell into the two segments representing the strongest relationship to science ("science connected" and "science enthusiast"), while just about ten percent of Sensorium participants belonged to the two top categories for science affinity. Overall, Sensorium participants exhibited a far weaker connection to science than in the comparison group of individuals who attended FIGMENT in general ($x2 = 32.7$, df = 4, p<0.001), The findings of a similar study at the Oregon Eclipse festival also showed that Guerilla Science participants came from the full range of audience segments, but science connected and science enthusiast were slightly more likely to opt into Guerilla Science activities. These results from FIGMENT support even more strongly than the results from the Oregon Eclipse Festival [19] the underlying hypothesis that Guerilla Science-style activities, like Sensorium, hold the potential to attract audiences to science experiences who are not strongly connected to science. Neither gender, age or education level affected the result of the relative representation by audience segmentation in our samples.

In order to understand whether Sensorium might have attracted a different type of audience for other reasons than interest in or affinity towards science, we compared data on visitor characteristics for Sensorium participant respondents to those of the comparison group. We found that Sensorium respondents were roughly equivalent to the comparison group overall

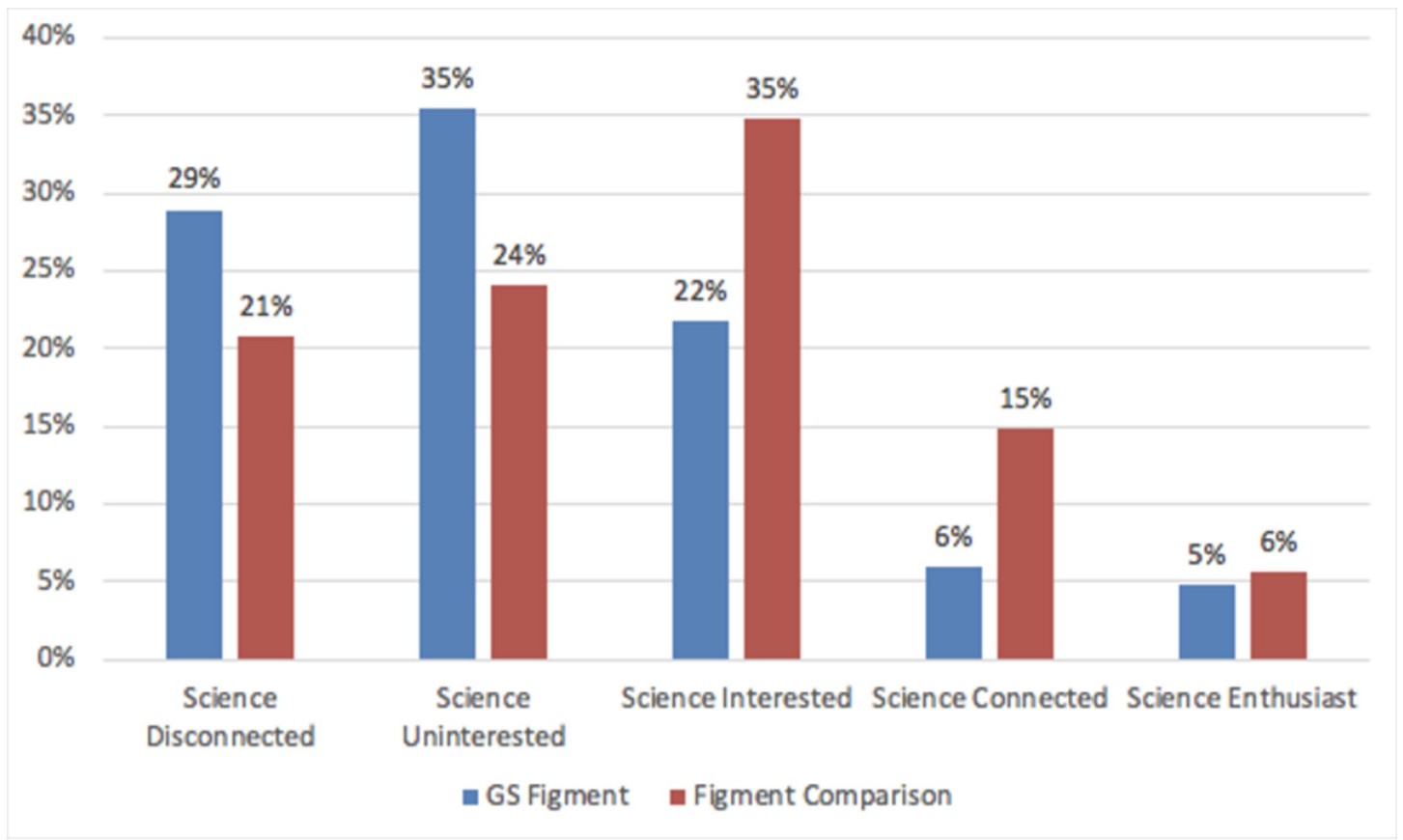

**Fig 6. Sample composition for five mutually exclusive audience segments by relationship to science from FIGMENT festival.**

in terms of age, even as they tended to be on the slightly younger side (Table 4 below). FIGMENT and Sensorium respondents were overwhelmingly young adults (defined as adults under 40): 65% of Sensorium sample, and 58% of the FIGMENT comparison sample were below 40. The middle age group (40–59) was equivalent between Sensorium and FIGMENT samples (33.3% vs 33.9%). The FIGMENT sample, though, included a larger proportion of adults 60 and older than the Sensorium one (7.8% vs 3.0%). More women than men attended FIGMENT (56% vs 43%) and Sensorium (59% vs 41%), and the difference between FIGMENT and Sensorium was negligible. Overall, the samples for Sensorium and the FIGMENT comparison group appear to be roughly equivalent.

### 3.3 How do FIGMENT visitors compare to a national sample on measures of science affinity?

Sensorium was able to attract an audience that was not skewed towards science, relative to FIGMENT overall. But what evidence do we have that FIGMENT overall was not attracting an audience with high affinity for science? That is, FIGMENT might itself not differ much from a science center or science café event in terms of its audience.

To give context to the data about participants at both FIGMENT overall and Sensorium, we asked Sensorium visitors about their prior engagement with in-person science-oriented and art-oriented spare-time activities. While national data suggest higher visitation to science-related places than art-related ones [8], Sensorium seems to have attracted an audience that is

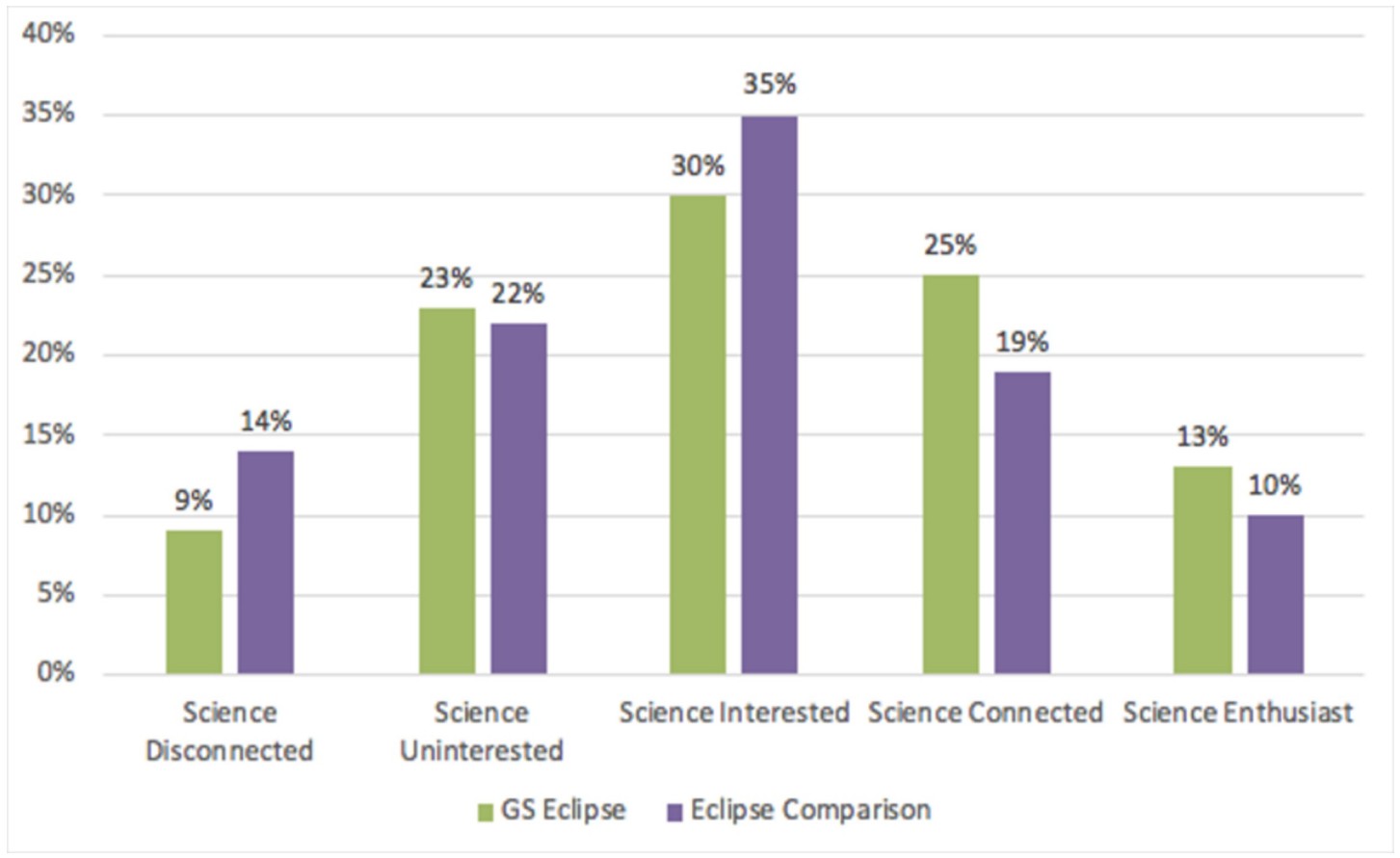

**Fig 7. Sample composition for five mutually exclusive audience segments by relationship to science from Oregon Eclipse festival, shown here for comparison with Fig 6.** Bisbee O'Connell et al., [19].

more art- than science-oriented with regard to their spare-time engagements: 82% of Sensorium participants visited an informal science institution at least once during the prior year, and 91% did so for art-related places (Table 5), in comparison to national data that reports 58% of the adult population visiting some kind of informal science institution at least once a year vs. 33% who visit art museums at least once a year [8]. Thus, our data indicate that Sensorium participants may not only have a much higher interest in all things cultural (whether they are art or science related), but show much stronger relative participation levels in art vs. science (roughly equal) to the relative participation levels in the general public. Thus, FIGMENT

Table 4. Age distribution of Sensorium and FIGMENT comparison samples.

| Age Group | Sensorium (n = 326) | Comparison (n = 448) |
|---|---|---|
| <21 | 2.5% | 2.0% |
| 21–29 | 28.7% | 21.0% |
| 30–39 | 33.6% | 35.3% |
| 40–49 | 22.7% | 22.1% |
| 50–59 | 9.6% | 11.8% |
| 60–69 | 2.2% | 5.6% |
| 70+ | 0.8% | 2.2% |

**Table 5. Prior engagement with art and science; only Sensorium participants (n = 232).**

| "Can you tell me how many times you have visited [. . .] in the past year, that is, the past 12 months?" | [zoo or aquarium, natural history museum, or science or technology museum] | [art museum, gallery, or concert] |
|---|---|---|
| I've gone at least once a month | 22% | 50% |
| I've gone every couple of months | 33% | 34% |
| I've gone at least once this year | 27% | 7% |
| It's been over a year since I've gone | 9% | 1% |
| I can't remember the last time I went | 6% | 4% |

festival-goers represent a slice of the population that generally engages in cultural activities and make use of cultural institutions in their environment. The data become even more dramatic when we look at higher frequency visits that may not show up in aggregate annual data (Table 5): 50% of Sensorium participants stated that they visit art-related spaces at least once a month vs. only 20% who visit informal science institutions once a month. That is, coarse national indicators as reported by NSB [8] might not capture the true story of the differences in the expression of art and science engagement in Sensorium participants. These finer-grained data suggest that Sensorium attracts an audience that, on average, attends science-related activities far less frequently than art-related events, and therefore express a much stronger affinity for art than for science, while also representing an audience that may visit science institutions more frequently than the average US resident. This result, describing so-called "culture vultures" is consistent with analogous surveys from the UK: "The segments who are most engaged with science [. . .] also appear to have greater non-science related cultural knowledge. This again suggests that there are perhaps not two cultures of science versus arts, but instead a group of people who are more engaged with both these areas" [38, p127]. Notwithstanding the high levels of cultural engagement within these groups, Sensorium was indeed able to attract to a science activity a segment that shows less personal affinity to science than the visitors to the festival overall.

## 3.4 What were the outcomes of participation in Sensorium at FIGMENT?

Overall, audience members were satisfied with Sensorium activities in which they participated. We measured satisfaction during exit interviews using a ten-point scale on three statements we later combined into a Net Promoter score: "I would recommend this event to a friend," "I would attend/take part in an event like this again," and "I am satisfied with this event." Individuals who score an average of 9 or higher are considered "Promoters": they had such a positive experience that they will tell others about it and actively recruit others to engage. Those with scores of 8 or 7 are considered "Passive": they were satisfied with their experience but will not become evangelists. Average scores of 6 and below indicate "Detractors," individuals who were unsatisfied with their experience and will share that dissatisfaction with others. Participants reported an average score of 8 on the combined scale and overall there were 58% Promoters and 12% Detractors, for a net promoter score of 46, indicating that participants in Sensorium on average had positive experiences.

To investigate the experience of participants we used closed-ended questions on Sensorium exit interviews and on short, written feedback forms within the officers' quarters, which constituted the main activity hub of Sensorium. Generally, respondents checked descriptors (of Sensorium events) at more than twice the rate on the feedback forms than during exit interviews, but the pattern that emerged remained similar: respondents considered the experience as fun, thought-provoking and informative. In contrast to the exit interviews, respondents also

characterized the experience as interactive on feedback forms. Negative associations such as confusing, intimidating or boring were chosen by fewer than 10% of respondents (Table 6).

Similar to event descriptors, exit interviews and written feedback forms included a closed-ended question around take-aways that provided a rough estimate on self-reported impact on participants (Table 7). Written feedback forms resulted again in about twice the rate of answers as exit interviews, and again a similar pattern emerged between both forms of audience feedback. A catch-all response item (I had a great experience) garnered the highest response, followed by perceived knowledge gain, and increased appreciation for science and for art. Few respondents did not experience any of the provided answer options.

A final estimate of impact of participation in Guerilla Science and Guerilla Science-style activities is based on behavioral intentions for follow-up activities as a result of participation in Sensorium, asked during exit interviews using Likert-type questions. Three-quarters of respondents expressed an interest in the topic or subject matter, although two related action items (looking up relevant information and following up with individuals or institutions) were endorsed by fewer respondents (Table 8). Almost four in five respondents were also interested in experiencing more Guerilla Science-style events, confirming other forms of expressions like satisfaction and event descriptors.

## 4. Discussion

The data supports the claim that the Guerilla Science's Sensorium was successful at broadening participation by engaging those visitors to FIGMENT and Governors Island who might otherwise not have engaged with science. We determined this by asking whether those who participated in Sensorium fundamentally differed from other visitors, particularly on characteristics that can be associated with affinity to science. Guerilla Science can therefore be considered a stealth or serendipitous form of science engagement which holds the potential for reaching and engaging new audiences.

We started the research on Guerilla Science with the hypothesis that two elements of science engagement might broaden participation in science for culturally-interested groups with low overall connection or relationship to science: (1) featuring science experiences at cultural events or settings that individuals will frequent for other reasons, and (2) designing experiences that blend elements of science with those of art and design. We hypothesized that the first feature addresses issues of *access* for those who would otherwise not seek out science, while the second feature addresses the issue of *inclusion*, by providing audiences with experiences that are not generally associated with typical science engagement at science festivals, science museums or similar type of informal science spaces and places [4]. We had investigated

**Table 6. Choice of descriptors for Sensorium events (check all that apply).**

|  | Percent Exit Interviews (n = 326) | Percent House Feedback Forms (n = 202) |
| --- | --- | --- |
| Fun | 45.7 | 75.7 |
| Thought-Provoking | 34.4 | 70.8 |
| Informative | 33.1 | 63.4 |
| Sociable | 20.6 | 27.7 |
| Inspiring | 14.1 | 32.2 |
| Interactive | 10.7 | 69.8 |
| Relevant | 9.2 | 21.8 |
| Confusing | 8.3 | 6.4 |
| Intimidating | 3.4 | 4.0 |
| Boring | 0.9 | 3.5 |

**Table 7. Take-aways from the experience at Sensorium (check all that apply).**

|  | Percent Exit Interviews (n = 326) | Percent House feedback forms (n = 202) |
|---|---|---|
| I had a great experience | 38% | 74% |
| I've gained new knowledge | 28% | 61% |
| I gained an increased appreciation for science | 23% | 50% |
| I gained an increased appreciation for art | 25% | 46% |
| I realized something about myself | 10% | 25% |
| I realized I can learn science | 8% | 19% |
| I experienced none of these things | 5% | 7% |

this effect in a previous study and found support for these hypotheses [19]. However, as noted above, the 2017 Oregon Eclipse festival was a multi-day and relatively costly music and art festival. Thus this study was designed both to confirm our initial findings in a different setting and to develop a more robust conceptual model for understanding the results.

Our new results at a very different event type broadly support these hypotheses again, and thereby confirm the previous findings about the potential of Guerilla Science for reaching culturally-engaged audiences. Attendees to Guerilla Science's Sensorium did not represent a selection of the FIGMENT visitors who skewed towards science; if anything, the opposite seemed to be the case. Self-selection into Sensorium based on higher affinity to science, therefore, did not occur, leaving us to conclude that Sensorium did not "cherry-pick" from the general visitorship to FIGMENT those with higher affinity to science. However, due to an overall well-above average participation in cultural institutions, including those which represent science, by visitors to FIGMENT, Sensorium might not have exposed new audiences to science (see Limitations section). Instead, it provided an engaging and novel approach towards blended science-art learning that the audience of a cultural festival appreciated and took part in. Consequently, Sensorium was rated and perceived positively by the majority of participants, and judging by expressed behavioral intentions, might hold the potential for inspiring subsequent reinforcing experiences [6].

Our results are most helpful when considered in the context of festivals as emancipatory spaces that invite participants to explore and inhabit new identities [26, 39]. The carnivalistic nature of the Sensorium was captured by audience responses, which most described as a fun but thought-provoking novel setting that they attended as an expression of their curiosity and desire to experience something new or different. How these experiences are to be modeled more generally as part of the participant's learning trajectory is a more nuanced question.

**Table 8. Take-aways from the experience at the Guerilla Science Sensorium (check all that apply).** Numbers represent rounded percentages (exit interviews, n = 326).

|  | Highly Disagree or disagree | Neither disagree or agree | Agree & Highly agree |
|---|---|---|---|
| I'd like to know more about the subject/topics I saw today | 5 | 19 | 77 |
| I'd like to come to more Guerilla Science Events | 4 | 12 | 75 |
| I'd look for event similar to Guerilla Science Events | 4 | 18 | 79 |
| I'd like to look up information on something I was introduced to at Guerilla Science today | 14 | 25 | 61 |
| I'd like to follow up with people or organizations I learned about at Guerilla Science today | 18 | 31 | 51 |

Instead of tackling this directly here, we suggest two potential models, each with their own advantages, and leave a more detailed investigation a topic for future research.

First, the *contextual model of learning* is a broad organizing framework that accounts for how the physical, socio-cultural, and personal contexts of the experience intersect, reinforce, and are reinforced by the participants' prior experiences and knowledge [6, 29, 39]. This model of learning in context has the advantage of being flexible in its applicability (it can be applied to almost any setting) and it is well known and accepted in informal education. However, in its focus on learning in one setting it is not naturally tailored to account for the impact of the type of "cultural interventions" studied here, primarily because it does not fully capture the comparative nature of the setting in which science or other cultural engagement might occur.

Second, *learning ecosystem model* is a model that captures how events and programs intersect with local civic, economic, and cultural ecosystems [40, 41]. The learning ecosystems model describes the "dynamic interaction among individual learners, diverse settings where learning occurs, and the community and culture in which they are embedded" [40 p.12]. At the conceptual center of the ecosystem model sits the learner, surrounded by and influenced by all the community's STEM-rich assets. Over time, and under the influence of the many dynamically interacting components of the ecosystem, learners can develop the skills and knowledge that are traditionally associated with STEM learning, as well as the broader socio-cultural facets of learning such as STEM identities, interest, and real-world understandings [42]. The advantage of this approach is that it naturally places events like the Sensorium within a broader social context for engagement and learning. In doing so, it provides a framework for considering their influence relative to, and in combination with those of other experiences. In an ecosystem approach, the role of a single intervention can, in theory, be untangled from a multitude of other factors, at the individual learner level [43]. Analogously, it was only when medieval carnivals were considered holistically within the broader constraints of society, could their role as a social safety valve be understood for the people—lords, clergy, and workers—that took part [28].

## 4.1 Limitations

Many discussions of broadening participation in science refer to economically disadvantaged and/or minority populations underrepresented in science. These populations often lack access to science engagement opportunities and may see little connecting them to typical settings for informal or free-choice science engagement [1, 10]. In this study, while we did address issues of access and inclusion, we did not investigate access and inclusion from the perspective of underrepresented ethnic or racial groups or those who lack the financial means to participate. Indeed, FIGMENT festival, despite being free to access and near public transit, still exhibited many recognized socio-cultural barriers to access, such as no dual-language website and a limited outreach program itself. Our study examined a far more narrow claim related to basic principles of free-choice learning, namely whether designing experiences that adhere to a learner's cultural norms and interests. While we were able to show that the Guerilla Science approach accomplished this, we are not able to confirm that Sensorium at FIGMENT reached new audiences for science. This limitation is entirely due to the unusually high rates with which even those who expressed little affinity to science already participate in informal science-related spare-time activities. Nonetheless, we can still see that Sensorium provided access to science for a large number of participants who rate their relationship to science as low.

Our data suggest that festival participants in New York represent an audience of particular high cultural engagement. Data from a 2019 survey conducted by Jon Miller (personal communication) with a representative sample of US adults suggest far lower frequencies for uses of

art- and science-related institutions in the US than in our New York City-focused sample from FIGMENT (Table 9). Miller finds low participation rates at a frequency of every couple of months or more, ranging from 3.3% for art galleries to 1.3% for planetariums. The data also suggest that a large portion of US adults do not engage with these cultural institutions at all, with a frequency of "none or can't recall" ranging from more than 80% for planetariums to 55% for zoos or aquariums. These comparisons again suggest, *assuming* that FIGMENT visitors are representative of New York in general, that our study reached into a segment of the U. S. population that is highly active in terms of cultural engagement. That is, our sample already contains individuals with low affinity to science who nonetheless frequent science-related educational institutions at relatively high rates.

Finally, while our underlying model invoked carnival and festivals as modes of creative and emancipatory production, our study did not empirically explore the mechanism by which the Guerilla Science intervention facilitated learning, or the specific role of the art-science integration. Although, related studies have shown that art-based approaches to engaging with science can lower hesitancy and open individuals to divergent thinking [44]. The underlying premise of this effect is that some learners may feel constrained by what they perceive science to stand for: being precise, requiring in-depth content knowledge and expertise, or requiring mathematical savvy. Framing science engagement through an art-based approach can free learners from potentially negative associations with science, thereby making it easier for them to engage. We also note that while our study may seem to fit neatly under the rubric of "STEAM" (STEM + Arts), and that many informal educators have embraced STEAM as an inclusive and authentic approach to engaging people with STEM, as a whole, we contend that the conceptualization and usage of STEAM is somewhat ambivalent and weakly theorized, as discussed in detail in [45]. We have, therefore, deliberately limited our descriptive language to avoid it.

## 5. Conclusion

Recent recommendations for overcoming historic inequities and dismantling structural barriers have accentuated the importance of taking an asset-based approach, valuing the experiences of diverse audiences, and recognizing that public(s) bring valuable knowledge and perspectives to the conversation, design elements that are central to the approach described here [2]. In this study we describe an evidence-based approach for engaging scientifically-underserved audiences which, we hope, can support both practitioners and researchers to

**Table 9. Frequency of use of arts and science institutions and resources, 2019 provided by Jon Miller (personal correspondence).**

| Institution | Frequency of use in % | | | |
|---|---|---|---|---|
| | At least once a month | Every couple of months | At least once this year | None or can't recall |
| **Arts institutions** | | | | |
| Art museum | 1.1 | 2.9 | 29.9 | 66.0 |
| Art gallery outside museum | 1.1 | 2.2 | 28.7 | 67.9 |
| Play or musical play | 0.9 | 4.0 | 37.0 | 58.0 |
| **Science institutions** | | | | |
| A natural history museum | 0.5 | 1.2 | 28.1 | 70.2 |
| A zoo or aquarium | 0.8 | 2.2 | 41.8 | 55.2 |
| A science center or museum | 0.7 | 1.1 | 35.2 | 63.0 |
| A botanical garden or arboretum | 0.8 | 1.8 | 32.6 | 64.9 |
| A planetarium | 0.3 | 1.0 | 16.3 | 82.5 |

N = 2,163

develop new tools and techniques to broaden participation. The primary value of this study, and that described in Bisbee O'Connell et al [19] is not, necessarily, that it provides guidance on reaching festival audiences and cultural consumers, although there is clearly intrinsic value in this guidance. Rather, its value is in advancing knowledge about how to recontextualize and implement established techniques, such as integrating science and art, in novel settings and with non-traditional audiences. (For example, the discussion earlier of approaching these projects as "carnival" is one way of recontextualizing these projects.) It is an open question to what extent this approach will need further modification in contexts or ecosystems where most attendees have a strong identity as part of a STEM-marginalized group such as women, racial minorities, veterans, or rural populations. No doubt the answer will depend heavily on the group in question, and is a subject for further research.

## Acknowledgments

Sincere thanks to our data collectors who spent many hours over the course of two days "interrupting" the flow of festival goers and therefore felt ongoing guilt about disrupting other people's fun spare-time activity. Thanks also to Kevin Keys for turning enormous stacks of handwritten data into electronic format.

## Author Contributions

**Conceptualization:** Mark S. Rosin, Martin Storksdieck, Kari O'Connell, Brianna Keys, Bruce V. Lewenstein.

**Data curation:** Martin Storksdieck, Kari O'Connell, Brianna Keys, Kelly Hoke.

**Formal analysis:** Martin Storksdieck, Kari O'Connell, Kelly Hoke.

**Funding acquisition:** Mark S. Rosin, Martin Storksdieck, Bruce V. Lewenstein.

**Investigation:** Kari O'Connell.

**Methodology:** Mark S. Rosin, Martin Storksdieck, Brianna Keys.

**Project administration:** Mark S. Rosin, Kari O'Connell.

**Resources:** Mark S. Rosin.

**Supervision:** Mark S. Rosin, Martin Storksdieck.

**Writing – original draft:** Mark S. Rosin, Martin Storksdieck.

**Writing – review & editing:** Mark S. Rosin, Martin Storksdieck, Kari O'Connell, Bruce V. Lewenstein.

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
