## [Decision Letter · Decision Letter 0]

7 Sep 2022

PONE-D-22-13920Broadening Participation in Science through Arts-Facilitated Experiences at a Cultural FestivalPLOS ONE

Dear Dr. Rosin,

Thank you for submitting your manuscript to PLOS ONE. After careful consideration, we feel that it has merit but does not fully meet PLOS ONE’s publication criteria as it currently stands. Therefore, we invite you to submit a revised version of the manuscript that addresses the points raised during the review process.

Please note that we have only been able to secure a single reviewer to assess your manuscript. We are issuing a decision on your manuscript at this point to prevent further delays in the evaluation of your manuscript. Please be aware that the editor who handles your revised manuscript might find it necessary to invite additional reviewers to assess this work once the revised manuscript is submitted. However, we will aim to proceed on the basis of this single review if possible. The reviewer was positive about your submission but has identified some opportunities to strengthen the conceptual grounding of the manuscript. We would ask that you revise the manuscript to address the considerations they have raised.

We look forward to receiving your revised manuscript.

Kind regards,

Jamie Males

Editorial Office

PLOS ONE

2. Please provide additional details regarding ethical approval and participant consent in the body of your manuscript. In the Methods section, please ensure that you have specified (1) the name of the IRB/ethics committee that approved your study, (2) whether consent was informed and (3) what type of consent you obtained (for instance, written or verbal). If your study included minors, state whether you obtained consent from parents or guardians. If the need for consent was waived by the ethics committee, please include this information.

“I have read the journal's policy and the authors of this manuscript have the following competing interests: Mark Rosin is a part owner of Guerilla Science Global and sits on its Board of Directors. The remaining authors declare that there are no conflicts of interest”

Reviewers' comments:

Reviewer's Responses to Questions

**Comments to the Author**

1. Is the manuscript technically sound, and do the data support the conclusions?

Reviewer #1: Yes

2. Has the statistical analysis been performed appropriately and rigorously? 

Reviewer #1: Yes

3. Have the authors made all data underlying the findings in their manuscript fully available?

Reviewer #1: Yes

4. Is the manuscript presented in an intelligible fashion and written in standard English?

Reviewer #1: Yes

5. Review Comments to the Author

Reviewer #1: Well-written, clear, and engaging, this article contains an in situ example of people's interaction with an art-framed, science-related experience. The step by step logic of this work results in a substantial scope of data, with testing of assumptions and alternative interpretations built into the design. Given this, my comments are minimal and largely conceptual in nature.

Whether these "serendipitous" (line 508) interactions produce longer-term science persistence or identity is an open question, as the authors are aware. The clarity of this manuscript shows that a broader audience can be reached through this type of approach, which supports pursuing follow-up questions about behavior, perception, and identity development for those not interested nor strictly expecting to engage in science. Yet, people's relationship to science and their science identity is likely more nuanced than the broad way it was measured here. The term "science" may be relatively fixed in people's minds, whereas their actual interaction with science is varied and potentially deep. For example, experience in building, cooking, or gardening involves many scientific processes although may not be viewed as "doing science" by someone by a strictly topic-based definition of science.

All this said, my comment is about whether the authors are hoping these types of events will help broaden how people define science (science can contain an element of art, culture, etc.) or whether it is about deepening someone's interest in "science" as a more fixed area of study. There are assumptions that the authors may be making about the line between science and art as two disciplinary practices, or set of topics, that I'd like to hear more about.

This breadth of defining science does not necessarily take-away from the findings here, as people's perception of "science" as a label is itself a motivating and filtering factor for pursuing science-related activities. Similarly, if there are confounding factors about how "science" is perceived, I do not have a reason to expect there to be a systematic difference between GS Figment and the comparison group. I also recognize there are practical considerations in assessing a more nuanced definition of science in a drop-in type environment--which is no small challenge!

6. PLOS authors have the option to publish the peer review history of their article (what does this mean?). If published, this will include your full peer review and any attached files.

Reviewer #1: No

---

## [Author Response · Author response to Decision Letter 0]

8 Nov 2022

RESPONSE: Confirmed. 

2. Please provide additional details regarding ethical approval and participant consent in the body of your manuscript. In the Methods section, please ensure that you have specified (1) the name of the IRB/ethics committee that approved your study, (2) whether consent was informed and (3) what type of consent you obtained (for instance, written or verbal). If your study included minors, state whether you obtained consent from parents or guardians. If the need for consent was waived by the ethics committee, please include this information.

RESPONSE: The Oregon State University (OSU) Institutional Review Board (IRB) approved the study as an exempt study. The OSU IRB approved a waiver of documentation (signature) of informed consent. For paper surveys, we included brief study information and consent language at the top of the survey, and participants gave their consent by filling out part or all of the paper survey and turning it into the box (labeled as such). For interviews, we read study information to potential participants before starting the interview, allowing for verbal consent. Our study only included adults (18+), and as part of the verbal consent we included the requirement to be 18 to be part of the study. 

Text to this effect now appears in section 2.3.

“I have read the journal's policy and the authors of this manuscript have the following competing interests: Mark Rosin is a part owner of Guerilla Science Global and sits on its Board of Directors. The remaining authors declare that there are no conflicts of interest”

RESPONSE: Confirmed. Statement is now added. See updated cover letter. 

RESPONSE: The data set on which this research is based is now publicly available at the following URL, hosted by the Scholars Archive at Oregon State University: https://doi.org/10.7267/hx11xq17s

A statement to this effect is now included in the cover letter. The data is also now cited directly in the paper. 

RESPONSE: The data set on which this research is based is now publicly available at the following URL, hosted by the Scholars Archive at Oregon State University: https://doi.org/10.7267/hx11xq17s

A statement to this effect is now included in the cover letter. The data is also now cited directly in the paper. 

RESPONSE: The updated text has been added to the manuscript in the penultimate paragraph to section 2.3.

Reviewers' comments:

Reviewer's Responses to Questions

Comments to the Author

5. Review Comments to the Author

Reviewer #1: Well-written, clear, and engaging, this article contains an in situ example of people's interaction with an art-framed, science-related experience. The step by step logic of this work results in a substantial scope of data, with testing of assumptions and alternative interpretations built into the design. Given this, my comments are minimal and largely conceptual in nature.

Whether these "serendipitous" (line 508) interactions produce longer-term science persistence or identity is an open question, as the authors are aware. The clarity of this manuscript shows that a broader audience can be reached through this type of approach, which supports pursuing follow-up questions about behavior, perception, and identity development for those not interested nor strictly expecting to engage in science. Yet, people's relationship to science and their science identity is likely more nuanced than the broad way it was measured here. The term "science" may be relatively fixed in people's minds, whereas their actual interaction with science is varied and potentially deep. For example, experience in building, cooking, or gardening involves many scientific processes although may not be viewed as "doing science" by someone by a strictly topic-based definition of science.

All this said, my comment is about whether the authors are hoping these types of events will help broaden how people define science (science can contain an element of art, culture, etc.) or whether it is about deepening someone's interest in "science" as a more fixed area of study. There are assumptions that the authors may be making about the line between science and art as two disciplinary practices, or set of topics, that I'd like to hear more about.

RESPONSE: 

This is an excellent question, and we thank the reviewer for raising it. 

We do indeed expect people’s relationship to science and their science identity to be more nuanced than the categorizations adopted in this article. Because the fieldwork took place in a loud, dynamic, busy environment in which survey participants had short attention spans, we chose, building on the associated research literature (see [Sec sec007], paragraph 1), to adopt the short non-intrusive questionnaires detailed in the Appendixes. It is an unfortunate feature of the kind of limited-survey, self-reporting empirical research conducted here that people’s interpretations of the questions, and the terms therein, such as “science” are inaccessible to us as researchers. (Furthermore, the same is true of the other interest and identity-based categories listed, for example, “politics”.)

Nevertheless, as the reviewer helpfully points out, it is important to make explicit in the article the associated underlying assumptions that follow from this fact. These assumptions, now stated in section 2.4, are that any variance in how the audience interprets terms like “science” will, on average, cancel out for a sufficiently large statistical sample. That is, on average, for every participant who interprets science in a narrow topic-based sense, another participant interprets it in a broader sense to include building, cooking, and gardening. In the absence of any systematic bias in the participants, this assumption is valid for the N>200 samples used here. 

One potential source of systematic bias in the data is suggested by earlier research [1] that showed that people who are engaging in authentic scientific research (through a citizen science process) do not identify their engagement as being “with science” unless they are explicitly told so. Nevertheless, in the absence of more concrete data to indicate this may be the case here, we retain our assumption above, while making it explicit by discussing this in the article directly. 

Text added in section 2.4: 

In conducting our statistical analysis, we ignore the possibility of any systematic bias in how participants interpret the term science, be it in a narrow topic-based sense or an expansive sense to include activities such as cooking or gardening. Earlier work has suggested that labeling an activity as "science" may have to be explicit for participants to change how they self-report their engagement with science [1]. 

[1] Brossard, Dominique, Lewenstein, Bruce V., & Bonney, Rick. (2005, 15 July). Scientific Knowledge and Attitude Change: The Impact of a Citizen Science Project. International Journal of Science Education, 27(9), 1099-1121.

---

## [Decision Letter · Decision Letter 1]

31 Jan 2023

PONE-D-22-13920R1Broadening Participation in Science through Arts-Facilitated Experiences at a Cultural FestivalPLOS ONE

Dear Dr. Rosin,

Thank you for submitting your manuscript to PLOS ONE. After careful consideration, we feel that it has merit but does not fully meet PLOS ONE’s publication criteria as it currently stands. Therefore, we invite you to submit a revised version of the manuscript that addresses the points raised during the review process.

The reviewers' comments are encouraging and I anticipate edits to be quite minor. Reviewer #3 confirmed the well-received clarity of the article. Reviewer #2 gave recommendations for emphasizing or foreshadowing the framework, contextualizing the current study among related efforts, and suggested small edits to wording and grammar. 

We look forward to receiving your revised manuscript.

Kind regards,

Meghan Bathgate, PhD

Guest Editor

PLOS ONE

Journal Requirements:

Reviewers' comments:

Reviewer's Responses to Questions

**Comments to the Author**

1. If the authors have adequately addressed your comments raised in a previous round of review and you feel that this manuscript is now acceptable for publication, you may indicate that here to bypass the “Comments to the Author” section, enter your conflict of interest statement in the “Confidential to Editor” section, and submit your "Accept" recommendation.

Reviewer #2: (No Response)

Reviewer #3: All comments have been addressed

2. Is the manuscript technically sound, and do the data support the conclusions?

Reviewer #2: Yes

Reviewer #3: Yes

3. Has the statistical analysis been performed appropriately and rigorously? 

Reviewer #2: Yes

Reviewer #3: Yes

4. Have the authors made all data underlying the findings in their manuscript fully available?

Reviewer #2: Yes

Reviewer #3: Yes

5. Is the manuscript presented in an intelligible fashion and written in standard English?

Reviewer #2: Yes

Reviewer #3: Yes

6. Review Comments to the Author

Reviewer #2: This is an interesting article that aims to broaden participation to science using a festival or carnival mode. The authors argue this is an emancipatory approach to learning, and also improves access and inclusion. While informal learning opportunities are valuable, many that are clearly science-identified may be missed by people who do not see themselves as science engaged, and therefore may miss out.

The main challenge I see in the piece right now is that the framework and “theory of change” is somehow scattered and embedded throughout the manuscript.

The authors would improve the manuscript by bringing the framework forward (or at least foreshadowing it) earlier on page 4. Specifically, the explanation on page 7 is compelling. However, I am trying to reconcile this with being “stealth” which is a word used in the research question. Stealth assumes trying to “hide” the intent of science, kind of like hiding veggies in a dish and surprising the person later that they ate veggies. Pop up implies unplanned, which can help people impromptu walk into something they might not otherwise intend to engage in.

Neither of these is as clearly “emancipatory” in the way a festival may promote community engagement and a broader invitation. This is revisited at bottom of page 28 and page 34 (carnival mode as emancipatory).

As a reader who is pretty familiar with the power of informal learning experiences, the authors need to put a sharper point on what’s new from this study…and to make it seem less like an evaluation of this activity. Toward this end, the discussion could put forward ideas of the kinds of exhibits or festivals that could help citizen science grab hold of our public.

The second suggestion is some kind of recognition that a STEAM movement already exists- so perhaps some brief contextualization about how guerilla science differs from STEAM. This is on page 4. There is also interesting data presented by the authors and by the personal communication of Jon Miller in Table 9 that suggest that science museum and art gallery attendees might be similar people. I think more could be made of this prospect, possibly in the introduction and then in the discussion.

Finally, small word choices could improve the readability of this article, particularly as the authors consider the opening of the article. There are some small word choices from the abstract through the first few pages that obscure the strength of this piece.

Abstract: Consider changing the word “choir” – in pages 3-4 you do a great job of explaining self-selection. Consider a word choice that brings forward the idea of self-selection. Those unfamiliar with choir metaphor may be confused by this in the abstract.

Page 3, Line: 36-37—That is, audiences who do not actively seek out science learning opportunities. This is a fragment- and should be connected to the previous sentence.

This challenge is motivated by two needs. Possibly rephrase: Addressing this challenge relies upon two principles

Page 20 line 379. One comma should be a period.

Reviewer #3: This study looks at the complicated setting of informal science learning to better understand design strategies for reaching broader audiences, defined by the authors as audiences with less access or inclusion in participating in science engagement.

The study design compares the general visitor to a regional arts festival to those who participate in a specific area of the festival- that incorporate stealth science experiences. It uses descriptive statistics to understand if the intervention reaches the desired broader audiences or if the experience results in self-selection by those with pre-existing science affinity. And builds on prior festival work, this time eliminating two access barriers (cost is free and public transportation is plentiful) while maintaining the stealth experiences that integrate arts and science expertise and content.

The authors cite broader impacts and inclusion as typically addressing race, income, ability, and gender but, in this study, broader audiences are visitors outside the average museum-going behavior. Data show that this arts festival in fact draws a higher than average group of cultural event visitors (science and art) but with more arts aligned than science aligned, yet all had positive outcomes from the stealth experience.

These experiences and this context were shown to be socially motivated, people came in groups, for a fun, active experience with an expectation to learn something, and in the stealth experience, these expectations were met and resulted in higher interest in science as part of that learning.

This study sheds light on our expectations about audiences in different contexts. The authors cite limitations to prior studies in terms of access, this study illuminates other assumptions about who attends a festival (compared to a museum or designated informal science learning context. Was this arts festival a draw to “culture vultures” more than other festivals might be?

Similar studies in an expanding variety of contexts for instance, county fairs; music or dance festivals centered in specific cultural communities; or local sports events where people gather to celebrate the effort of community members, each could have similar stealth experiences built in that teach us how people connect their local activities to science, art and social goals and questions.

This is a well-designed and written study that informs both festival and informal learning institutions in thinking about expectations and motivations as they attempt to engage visitors in science through different frames.

7. PLOS authors have the option to publish the peer review history of their article (what does this mean?). If published, this will include your full peer review and any attached files.

Reviewer #2: No

Reviewer #3: No

---

## [Author Response · Author response to Decision Letter 1]

16 Mar 2023

Our response to the reviewers is attached as a formatted word document.

---

## [Editor Report · Decision Letter 2]

3 Apr 2023

Broadening Participation in Science through Arts-Facilitated Experiences at a Cultural Festival

PONE-D-22-13920R2

Dear Dr. Rosin,

We’re pleased to inform you that your manuscript has been judged scientifically suitable for publication and will be formally accepted for publication once it meets all outstanding technical requirements.

Kind regards,

Meghan Bathgate, PhD

Guest Editor

PLOS ONE

Additional Editor Comments (optional):

The authors have thoughtfully and appropriately addressed the comments and questions raised by the reviewers. Thank you for sharing this work!
---

## [Editor Report · Acceptance letter]

28 Apr 2023

PONE-D-22-13920R2 

Broadening Participation in Science through Arts-Facilitated Experiences at a Cultural Festival 

Dear Dr. Rosin:

I'm pleased to inform you that your manuscript has been deemed suitable for publication in PLOS ONE. Congratulations! Your manuscript is now with our production department. 

Kind regards, 

on behalf of

Dr. Meghan Bathgate 

Guest Editor

PLOS ONE